# Salinity Alleviation and Reduction in Oxidative Stress by Endophytic and Rhizospheric Microbes in Two Rice Cultivars

**DOI:** 10.3390/plants12050976

**Published:** 2023-02-21

**Authors:** Amrita Gupta, Arvind Nath Singh, Rajesh Kumar Tiwari, Pramod Kumar Sahu, Jagriti Yadav, Alok Kumar Srivastava, Sanjay Kumar

**Affiliations:** 1Amity Institute of Biotechnology, Amity University Uttar Pradesh, Lucknow Campus, Lucknow 226028, UP, India; 2ICAR-Indian Institute of Seed Sciences, Kushmaur, Maunath Bhanjan 275103, UP, India; 3ICAR-National Bureau of Agriculturally Important Microorganisms, Kushmaur, Maunath Bhanjan 275103, UP, India

**Keywords:** endophytes, rhizospheric microbes, salinity, root architecture, *Bacillus*, oxidative stress, reactive oxygen species, Na^+^/K^+^ balance, Sodium Green

## Abstract

Increased soil salinity poses serious limitations in crop yield and quality; thus, an attempt was made to explore microbial agents to mitigate the ill effects of salinity in rice. The hypothesis was mapping of microbial induction of stress tolerance in rice. Since the rhizosphere and endosphere are two different functional niches directly affected by salinity, it could be very crucial to evaluate them for salinity alleviation. In this experiment, endophytic and rhizospheric microbes were tested for differences in salinity stress alleviation traits in two rice cultivars, CO51 and PB1. Two endophytic bacteria, *Bacillus haynesii* 2P2 and *Bacillus safensis* BTL5, were tested with two rhizospheric bacteria, *Brevibacterium frigoritolerans* W19 and *Pseudomonas fluorescens* 1001, under elevated salinity (200 mM NaCl) along with *Trichoderma viride* as an inoculated check. The pot study indicated towards the presence of variable salinity mitigation mechanisms among these strains. Improvement in the photosynthetic machinery was also recorded. These inoculants were evaluated for the induction of antioxidant enzymes viz. CAT, SOD, PO, PPO, APX, and PAL activity along with the effect on proline levels. Modulation of the expression of salt stress responsive genes *OsPIP*1, *MnSOD*1, *cAPXa*, *CATa*, *SERF*, and *DHN* was assessed. Root architecture parameters viz. cumulative length of total root, projection area, average diameter, surface area, root volume, fractal dimension, number of tips, and forks were studied. Confocal scanning laser microscopy indicated accumulation of Na^+^ in leaves using cell impermeant Sodium Green™, Tetra (Tetramethylammonium) Salt. It was found that each of these parameters were induced differentially by endophytic bacteria, rhizospheric bacteria, and fungus, indicating different paths to complement one ultimate plant function. The biomass accumulation and number of effective tillers were highest in T4 (*Bacillus haynesii* 2P2) plants in both cultivars and showed the possibility of cultivar specific consortium. These strains and their mechanisms could form the basis for further evaluating microbial strains for climate-resilient agriculture.

## 1. Introduction

Climate change poses a great challenge worldwide for food security, which is an area with high-priority on the list of UN Sustainable Development Goals [1]. Increased abiotic stress is one of the greatest challenges arising out of climate change. Several abiotic stresses are hampering the quality and quantity of the produce [2]. Salinity is one of the abiotic stressors that is severely hampering crop growth, and the effect is increasing day-by-day. Soil salinity causes serious limitations in achieving the yield potential of a cultivar. Increasing soil salinity is a grave threat to the crop production system. There is a sharp decline in both quality and quantity of produce due to increasing soil salinity. Intensive agricultural practices have made the expansion of saline soil faster. Salinity stress in plants imparts serious ill effects on nutrient uptake, osmotic balance, membrane integrity, and overall growth, thus hampering the whole crop dynamics [2]. It also causes generation of excessive reactive oxygen species, which besides acting as signaling molecule, it can harm plant function and reduce productivity at higher concentrations [3]. A large acreage of quality land is coming under salinity every year. This poses serious limitations to crop productivity and limits sustainable land use. Any attempt of reducing the effects of salinity in a plant system that could support improved growth under elevated salinity could be an important strategy for developing climate-resilient agriculture.

Microbes are reported to have a close association with plants for nutrient cycling and alleviation of biotic and abiotic stresses [4,5]. Microbes suitable for mitigating the deleterious effects of soil salinity on plant growth and productivity are being explored for sustainable agriculture. Microbes are reported to have a tremendous capacity to sustain plant growth under salinity, such as improving nutrient uptake, osmotic balance, ionic balance, membrane stability, overall growth, etc. [6]. The varietal development is although an option for developing climate-resilient cultivars, but has limitations of finding tolerant donors in each crop. Whereas in the case of salinity alleviating microbes, it can be applied to any rolling varieties of a number of crops. Exploring endophytes and rhizospheric microbes that could improve such ill effects of salinity on different physiological parameters could be a great resource in crop package and practices. Induced systemic tolerance is reported as one of the most crucial mechanisms by which microbes help plants in mitigating the effects of salinity [7]. Plants have antioxidant enzymes that guard them against the damaging effects of extremely high reactive oxygen species (ROS) created during stress, which are induced from microbial inoculation [8].

Microbes have a vast functional diversity [9]. They can perform salinity alleviation as rhizosphere microbes or as endophytes. Rhizosphere microbes act on the rhizosphere and could be instrumental in the plant-soil interface where plants encounter salinity, and endophytic microbes act inside the plant system where the ill effects of salinity are realized [5]. Therefore, exploring the combined possibility of screening rhizosphere and endophytes microbes for conferring stress tolerance could provide the benefits of two different niches, which could be used in complementing each other as inoculants. It is again important to evaluate that whether the difference in mechanisms of rhizospheric and endophytic microbes could bring significant changes in crop growth under salinity.

In the agriculture sector, rice (*Oryza sativa* L.) is considered as staple food for billions of mouths. Since rice cultivation is undertaken in flooded conditions, a huge amount of salt is accumulated in the upper soil layer as the water evaporates; this soil salinity affects the crop development. Thus, rice could be considered as a model system to study salt alleviation. For our study, we have taken two cultivars with distinct features. The CO51, which is a short-duration, high-yielding rice cultivar, and has a higher tolerance to stresses [10]. The second variety was Pusa Basmati 1 (PB1), which is the world’s first semi-dwarf Basmati variety, has higher yields, and is the most widely grown Basmati variety, but is relatively susceptible to some stresses [11]. Looking at the increasing detrimental effects of salinity, it is important to characterize different microbial systems for staple food crops, such as rice, so that the base for effective climate-resilient cropping strategies could be widened and make Indian farming future-ready. Therefore, it was considered worthwhile to evaluate the potential of endophytic and rhizospheric bacteria in the mitigation of salt stress in rice. With this objective and during this period, a pot trial of two rice varieties, Pusa Basmati-1 (PB1) and CO51, was conducted with two endophytes, two rhizospheric bacteria, and a *Trichoderma* strain as the standard inoculation.

## 2. Results

The halotolerant endophytes and rhizobacteria were screened *in planta* and were found to influenece the dry matter accumulation, root-shoot length, increased antioxidant activities, and supplemented plants’ machinery for abiotic stress mitigation. *Bacillus safensis* BTL5, *Bacillus haynesii* 2P2, *Brevibacterium frigoritolerans* W19, and *Pseudomonas fluorescens* 1001 were effective in vitro as well as in the *in planta* trial (Figure 1). *Trichoderma viride* was taken as the standard inoculated check. The effect of salinity and inoculation were visible in plant growth and development (Figure 2). Significant differences were found in root development and corresponding differences could be observed from shoot growth.

### 2.1. Chlorophyll and Carotenoids Content

In CO51, highest chlorophyll content was obtained from T6 (2.06 mg g^−1^ FW), while T2 plants had a significantly lower (1.19 mg g^−1^ FW) chlorophyll content (Figure 3A). Whereas in PB1, the highest chlorophyll content was obtained from T4 and T3 (2.39 and 2.32 mg g^−1^ FW, respectively), while T2 plants recorded a significantly lower (1.10 mg g^−1^ FW) chlorophyll content. In both CO51 and PB1, the carotenoid content was highest in T5 plants (0.78 and 0.89 mg g^−1^ FW, respectively) and lowest was observed in T2 plants (0.31 and 0.44 mg g^−1^ FW, respectively; Figure 3B).

### 2.2. Proline Content

In CO51, the highest proline accumulation was found in T6 (3.04 µmol g^−1^ FW) and T4 (2.87 µmol g^−1^ FW), while T1 plants had a significantly lower (1.24 µmol g^−1^ FW) accumulation (Figure 3C). In the case of PB1, the highest chlorophyll content was obtained from T3 and T6 (2.25 and 2.18 µmol g^−1^ FW, respectively), and T1 plants recorded a significantly lower (1.05 µmol g^−1^ FW) proline content.

### 2.3. Antioxidant Enzymes

In CO51, the highest CAT activity was found in T4 (3364.92 µmol ml^−1^), whereas plants in T1, T3, and T7 treatments had significantly lower CAT activity. In the case of PB1, the highest CAT activity was obtained from T7 (3374.35 µmol ml^−1^), whereas plants in T1, T4, and T5 treatments had the lowest CAT activity (Figure 4A).

In CO51, the highest SOD activity was found in T3 (8.25 Unit g^−1^ FW), whereas plants in T1 and T5 treatments had significantly lower SOD activity. In the case of PB1, the highest SOD activity was obtained from T4 and T6 (8.21 and 7.97 Unit g^−1^ FW^1^, respectively), whereas plants in T1 had the least SOD activity (2.54 Unit g^−1^ FW; Figure 4B).

In CO51, the highest PO activity was found in T5 (3.69 Unit min^−1^ g^−1^ FW), and T1 plants had a significantly lower (2.2 Unit min^−1^ g^−1^ FW) accumulation. In the case of PB1, the highest PO activity was obtained from T3 (3.42 Unit min^−1^ g^−1^ FW), and T1 plants recorded significantly lower (2.18 Unit min^−1^ g^−1^ FW) PO activity (Figure 4C).

In CO51, the highest PPO activity was found in T5 and T4 (1.33 and 1.28 ∆O.D. min^−1^ mg^−1^ FW, respectively), whereas plants in T1 (0.49 ∆O.D. min^−1^ mg^−1^ FW) had significantly lower PPO activity. In the case of PB1, the highest PPO activity was obtained in T5 and T4 (1.29 and 1.28 ∆O.D. min^−1^ mg^−1^ FW, respectively), whereas plants in T1 had the lowest PPO activity (0.71 ∆O.D. min^−1^ mg^−1^ FW), which was on par with T3 and T6 (0.73 and 0.78 ∆O.D. min^−1^ mg^−1^ FW, respectively; Figure 4D).

In CO51, the highest ascorbate peroxidase (APx) activity was found in T2 (4.72 µmol g^−1^ FW) followed by T6 (4.5 µmol g^−1^ FW), whereas T1 plants had significantly lower APx activity (3.3 µmol g^−1^ FW). In the case of PB1, the highest APx activity was obtained from T7 and T6 (4.18 and 4.11 µmol g^−1^ FW, respectively), whereas T1 and T5 treatment had the lowest APx activity (2.15 and 2.30 µmol g^−1^ FW, respectively; Figure 4E).

In CO51, the highest PAL activity was found in T6 (1641.43 µmol TCA g^−1^ FW), whereas plants in T1 (1426.27 µmol TCA g^−1^ FW) had significantly lower PAL activity, which was on par with T7 (1470.16 µmol TCA g^−1^ FW). In the case of PB1, the highest PAL activity was obtained from T4 and T5 (1722.48 and 1751.07 µmol TCA g^−1^ FW, respectively), whereas plants in T1 had the lowest PAL activity (1488.94 µmol TCA g^−1^ FW; Figure 4F).

### 2.4. Shoot and Root Dry Weight

In CO51, the highest shoot dry weight was found in T5 (8.19 g), which was on par with T4 (8.10 g), while T1 plants had a significantly lower shoot dry weight (6.63 g; Table 1). In PB1, the highest shoot dry weight was found in T4 (12.08 g), while T1 plants had a significantly lower shoot dry weight (7.39 g). In CO51, the highest root dry weight was found in T4 (6.60 g), which was on par with T5 (6.27 g) and T7 (6.26 g), while T1 plants had a significantly lower dry weight (4.97 g). In PB1, the highest root dry weight was found in T4 (7.99 g), which was on par with T5 (7.61 g), while T1 plants had a significantly lower root dry weight (6.45 g).

### 2.5. Shoot and Root Length

In CO51, the highest shoot length was found in T6 (84 cm), while T7, T3, and T1 plants had a significantly lower shoot length (Table 1). In PB1, the highest shoot length was found in T6 (74 cm), which was on par with T2, T3, T5, and T7, while T1 plants had a significantly lower shoot length (68.33 cm). In CO51, the highest root length was found in T1 (34.33 cm), which was on par with T4 and T7, while T6 plants had a significantly lower root length (22.33 cm), which was on par with T2 and T3. In PB1, the highest root length was found in T3 (43 cm), while T7 plants had a significantly lower root length (33.67 cm).

### 2.6. Number of Tillers

In CO51, the highest number of tillers was found in T7 (8.92), while T2 plants had a significantly lower number of tillers (3.92) (Table 2). In PB1, the highest number of tillers was found in T7 (7.75), which was on par with T1 and T6, while T2 plants had a significantly lower number of tillers (5.62). In CO51, the number of effective tillers was found to be highest in T7 (8.50), while T2 plants had a significantly lower number of effective tillers (3.67). In PB1, the highest number of effective tillers was found in T6 (6.17), which was on par with T1, T4, and T5, while T2 plants had a significantly lower number of effective tillers (3.95).

### 2.7. Root Parameters

In the case of CO51, the cumulative length of the total root present in the entire root system was significantly highest in the negative control (2861.81 cm) and T7 (2931.01) treated plants (Table 3). The projection area, surface area, and number of forks were highest in the negative control followed by T7. Average diameter, root volume, and fractal dimension were highest in T5 plants. The highest number of tips was found in T7. In PB1, the cumulative length, projection area, surface area, average diameter, fractal dimension, and number of tips and forks were highest in T7 followed by T4. The root volume was highest in T7 and T4.

### 2.8. Gene Expression Study

In this study, we have assessed the expression of the *OsPIP*1, *MnSOD*1, *cAPXa*, *CATa*, *SERF*, and *DHN* genes (Figure 5). Changes in the expression of salt stress responsive genes were recorded from the treatments inoculated with microbial agents. In this study, both cultivars resulted in differential gene expression. In CO51, the expression of the *OsPIP*1 gene was highest in *Bacillus safensis* BTL5 (T5) and *Brevibacterium frigoritolerans* W19 (T6) inoculated plants, whereas inoculation of *Trichoderma viride* was found to downregulate *OsPIP*1 expression. However, in PB1, *Bacillus safensis* BTL5 (T5) resulted in the highest expression, followed by the positive control (T2) and *Pseudomonas fluorescens* (T7). In the case of *MnSOD*1, CO51 plants showed the highest expression in T6 and T3, whereas PB1 plants had the highest expression in T6 (*Brevibacterium frigoritolerans*). The *cAPXa* gene expression was upregulated the most by T6 and T5 in CO51, and T7 and T6 in PB1. In CO51, *CATa* gene expression was highest in T4 and T5 plants, whereas in PB1, the highest fold change was recorded from T7 and T5 plants. In the case of the *SERF* transcription factor, CO51 plants recorded the highest fold increase in T5 and T3, whereas PB plants had the highest fold increase in T6 plants. The *DHN* gene expression was highest among CO51 plants in the T7 treatment, whereas T5 had the highest fold change among PB1 plants.

### 2.9. CSLM Study for Na^+^ Accumulation in Leaf Tissues

In this study, a relative higher Na^+^ accumulation was seen in the leaves of both cultivars. In CO51, the higher sodium accumulation was seen from the positive control and was less in all inoculated treatments (Figure 6). In the positive control, the accumulation was higher between sclerenchyma cells, whereas among other treatments, the accumulation was less between sclerenchyma cells. T4 and T7 had a sodium green fluorescence similar to the negative control. In PB1, the positive control had the highest sodium green fluorescence, followed by T3, T4, and T5. The lowest accumulation was seen from the negative control, T4, and T7 treatments.

## 3. Discussion

Several microbial agents have been reported to have protective roles under elevated salinity [6]. Microorganisms have been found to re-establish ion homeostasis and reduce the ill effects of ion toxicity and oxidative stress. In this study, the potential of endophytic and rhizosphere bacteria was tested in two different rice cultivars for salt stress alleviation under pot conditions.

In the present study, the decrease in the biomass accumulation (root and shoot dry weight) and number of tillers from the addition of 200 mM additional NaCl in T2 could be seen as an ill effect of salinity as compared to the non-positive control (T1). Cells maintain a certain balance among different ions for cell functioning. Excess salt in soil solution cause ion imbalance resulting in ion toxicity in the cell, which disturbs normal cellular processes [5]. As a side effect, an excess of reactive oxygen species is generated in plant cells under stress. This leads to reduced growth and biomass accumulation [6]. The inoculation of microorganisms (treatments T3–T7) under similar saline conditions to T2 could improve biomass accumulation and the number of tillers (Table 2). The increase in tillers is directly associated with the yield of rice. The results showed that, in both cultivars, inoculated treatments had a significantly higher number of effective tillers. This could be due to higher photosynthetic machinery in inoculated treatments. Our results on the chlorophyll and carotenoid content showed them to be higher in all inoculated conditions than the positive control in both cultivars. High salinity affects the photosynthetic pigments and reduces the photosynthetic rate, as seen from the chlorophyll content of positive control plants (T2). It has been reported, in other studies, that the application of endophytic and rhizospheric bacteria could improve chlorophyll accumulation [6]. This could be one of the direct reasons for the increased dry matter accumulation. An increase in chlorophyll and carotenoids by microbial inoculation has been documented previously. This increase in photosynthetic machinery is directly related to the dry matter accumulation.

Alleviation of ion toxicity and oxidative stress from plants would be a reason for higher photosynthetic pigments. Inoculation of salinity tolerance microbes are reported to reduce the accumulation of Na^+^ ions and improve the ionic balance of cells, as seen in Figure 6, where confocal scanning laser microscopy showed the reduced accumulation of Na^+^ ions in inoculated plants under salinity. A similar report was shown by Sahu et al. [6] in tomatoes under salinity with possible explanations of Na^+^ exclusion, uptake of K^+^, compartmentalization in vacuoles, etc., for the reduced Na^+^ content. Plant tissues with different NaCl applications showed a similar difference in Sodium Green™ Tetra (Tetramethylammonium) fluorescence [12]. This Sodium Green™ represents light-excitable Na^+^ indicator fluorescent probes, which gives information regarding Na^+^ concentrations and has a very high specificity as compared to other monovalent cations, such as K^+^ [13].

This reduction in Na^+^ accumulation could be due to the activation of salinity stress responsive regulatory genes *SERF1* (Figure 5E) and *DHN* (Figure 5F), which activate multiple pathways for salinity tolerance. SERF1 (SALT-RESPONSIVE ERF1) activates other transcription factors responsible for salinity stress mitigation such as MAPKs, DREB2A, and Zinc Finger Protein [14]. It also regulates ROS-mediated salinity stress signaling [15]. DHN gene is key regulator for abiotic stress responses in plants, having important role in scavenging excess ROS, and has been reported in rice to alleviate salinity stress [16].

Increased photosynthetes translocation to the roots could have additionally helped plants to respond to microbial signals for improved root growth and to develop a robust root system to fight salinity. Table 2 shows the improvements in different root architecture parameters by microbial inoculation. Roots are highly affected by excessive salinity [17,18]. Root architecture improvement could help plants to survive in higher soil salinity [19]; such improvements in parameters including the root projection area, root volume, surface area, number of lateral roots, and number of forks and tips in inoculated plants could have helped rice plants to absorb nutrients and water. As in our study, modulation in the expression of the hydroporins gene *OsPIP*1 in the roots was found in T5 and other treatments (Figure 5A). Similar to these results on root architecture improvement by inoculation, several other workers also reported the relevance of root parameters with salinity alleviation [20,21,22]. Secretion and regulation of plant hormones by these microbes could be very important mechanisms behind influencing root development [23,24,25]. This study provides details on the different root architecture parameters in CO51 and PB cultivars of rice under elevated salinity.

The ability of endophytic and rhizospheric bacteria to reduce the effects of high salinity could have resulted in increased biomass accumulation under 200 mM NaCl stress. Sahu et al. [5,6] have discussed several mechanisms by which these microbes could improve plant metabolism under salinity stress. One of them is the accumulation of compatible solutes such as proline. The proline accumulation study (Figure 5) showed higher accumulation in the plants treated with potential microbes. In the CO51 cultivar, T6 and T4 plants had the highest proline accumulation, whereas in PB1, T6 and T3 plants had higher proline accumulation. This increase in proline accumulation could be partially responsible for the improved ion homeostasis and reduced osmotic imbalance, which is in line with the reports of Das and Roychoudhury [26], suggesting that proline is responsible for minimizing the harmful effects of ROS under stress. Similarly, the application of *Staphylococcus haemolyticus* and *Bacillus subtilis* improved the production of different osmolytes, such as proline, which contributed to enhancing plant performance under salinity [27]. Proline is also reported to induce the expression of several other stress responsive genes in the plants [28]. The reports of Sahu et al. [6] and Nguyen et al. [29] indicate that proline has a dual role as an antioxidant and in osmoregulation in salinity stress alleviation. Some reports suggested that inoculation of bacterial agents upregulated the expression of genes responsible for the biosynthesis of proline [30]. Increased accumulation of compatible solutes is reported as an important salt stress alleviation strategy [5]. It can effectively reduce the ill effects of salinity and allow the cells to maintain cellular homeostasis.

Similar results were observed from our study regarding the modulation of the antioxidant enzymes (superoxide dismutase, catalase, peroxidase, phenylalanine ammonia lyase, and polyphenol oxidase) from the plants inoculated with different microbes. Though these antioxidant enzymes were differentially activated in both cultivars by the applied microbes, they were all improved under salinity stress. There are numerous instances of endophytes enhancing plants’ antioxidant enzymatic activity as a crucial method for reducing salt stress [6]. However, the pattern in our study was varying among treatments and cultivars (Figure 4). In the case of peroxidase, the T5 inoculation had the highest accumulation in CO51, whereas *Trichoderma* inoculation showed a higher accumulation in PB1. In catalase, CO51 had the highest accumulation by T4 inoculation but PB1 had the highest accumulation by T7 inoculation. This could be due to varied mechanisms of the inoculants for induced systemic tolerance. Some other studies also indicated that distinct microbial inoculants have diverse strategies for reducing salt stress [6]. This also indicates towards complementing roles of different microbial inoculants for the fitness of host plants. As a holobiome, the host plant is in active interaction with numerous microbes. It may be possible that all these interactions could yield a different response in the plant system. A similar response was recorded from our study; all five antioxidant enzymes, superoxide dismutase, catalase, peroxidase, phenylalanine ammonia lyase, and polyphenol oxidase, are activated differently in both of the cultivars. These enzymes may be a rationale for the decreased ROS damage in rice plants, including the activation of PO, CAT, PPO, PAL, and SOD activities through the induction of systemic tolerance. The gene expression study also validated the modulation of genes for antioxidant enzymes biosynthesis (*CATa*, *cAPX*, *MnSOD*1) in both CO51 and PB1. A plant’s antioxidant machinery being supplemented by microbial inoculation would be more economically significant for reducing the negative impacts of salt stress. The performance is different in both cultivars, which shows that both cultivars respond differently to the microbial inoculation. This is similar to the findings of Sahu et al. [9], where differential microbial functions were reported from two different cultivars.

## 4. Materials and Methods

### 4.1. Preparation of Pots, Inoculation and Transplanting

Pots were filled with 5 kg of non-sterile field soil and farm-yard-manure (FYM) in a 3:1 ratio. A blanket application of NPK fertilizer (@120:80:40 kg ha^−1^) was also applied as the basal dose. Row and column randomization was done thrice to avoid any heterogeneity in soil. After randomization, three wetting and drying cycles were given to bring the soil to natural compaction. In this study, seven treatments were given in two different rice cultivars, CO51 and PB1, with three replications of each. The nursery was raised for these two cultivars as per the protocol given in Nawaz et al. [31]. The microbial inoculation was done by seedling dip in respective treatments (Table 4). Seedlings were in respective culture broths (2 mL per liter) for a period of 30 min with 0.01% carboxymethyl cellulose as a sticking agent. After a period of 1 h, the seedlings were transplanted into the pots. Two paddy seedlings per hill, having equal height, and two hills per pot were transplanted in the pots. Plants were raised following standard cultivation practices and the observations were taken time to time. In brief, rice plants were irrigated at field capacity, and 50% of nitrogen was applied in two split doses in a 30-day interval after inter-culture operation. Pots were randomized twice during the growth period to avoid any heterogeneity in light interception.

### 4.2. Chlorophyll and Carotenoids Content

Leaves were taken at 60 days after transplanting to assess the chlorophyll and carotenoids content, as described by Witham et al. [32]. Briefly, one gram of leaf tissue was crushed in 80% pre-chilled acetone and the volume was made up to 100 mL with pre-chilled acetone. The absorbance of the supernatant was recorded at 452, 663, and 645 nm using a UV–vis 1700 spectrophotometer, Shimadzu, Japan. The amount of chlorophyll and carotenoids present (mg/g) in leaf tissue was calculated by the formula given in Sadashivam and Manickam [33].

### 4.3. Proline Content

Proline content was measured by crushing 0.5 g sample in 10 mL 3% aqueous sulphosalicylic acid followed by filtering with Whatman no. 2 filter paper. In 2 mL of filterate, 2 mL glacial acetic acid and 2 mL acid ninhydrin was added and kept in a boiling water bath for a period of 1 h. The reaction was terminated by placing it in an ice bath. Further, toluene (4 mL) was mixed by stirring for 20–30 s, and the solution was kept at room temperature for toluene layer separation. The upper layer is taken, and the absorbance of the red color was taken at 520 nm with a UV–vis 1700 spectrophotometer, Shimadzu, Japan. Calculations were performed using a standard curve, as described in Sadashivam and Manickam [33].

### 4.4. Electrolyte Leakage

The leakage of electrolytes from rice leaves was assessed by the autoclaving method. Ten leaf discs were taken from both cultivars from different treatments, placed in 25 mL deionized water, and incubated at ambient temperature for 4 h. Post incubation, the content was autoclaved at 121 °C for 30 min. The electrical conductivity was measured before and after autoclaving and electrolyte leakage was calculated as per Khare et al. [34].

### 4.5. Antioxidant Enzymes

#### 4.5.1. Peroxidase (PO)

The enzyme extract was prepared using 1 g fresh plant tissue, which was ground in phosphate buffer (0.1 M, 3 mL, pH 7.0) using a pestle and mortar. The content was centrifuged at 12,000 rpm for 15 min. The 100 µL enzyme extract was mixed with 50 μL of 20 mM guaiacol solution and 3 mL 50 mM phosphate buffer. Finally, 30 μL of 12.3 mM H_2_O_2_ was added to the cuvette to start the reaction and absorbance was recorded at 436 nm using a UV–vis 1700 spectrophotometer, and ∆t was calculated (Hammerschmidt et al. [35]).

#### 4.5.2. Catalase (CAT)

Catalase activity was measured as per the protocol followed by Luck [36]. Tissue homogenization and preparation of the enzyme extract was followed similarly to the peroxidase. The reaction mixture was prepared by mixing 100 µL enzyme extract and 3 mL 50 mM phosphate buffer. The addition of 30 μL of 12.3 mM H_2_O_2_ was done at the last step in the cuvette to start the reaction. The H_2_O_2_ degradation was recorded at 240 nm using a UV–vis 1700 spectrophotometer, and ∆t was calculated.

#### 4.5.3. Superoxide Dismutase (SOD)

SOD activity was measured as per the protocol followed by Beauchamp and Fridovich [37]. The enzyme extract was prepared using 1 g fresh plant tissue, which was ground in phosphate buffer (0.1 M, 3 mL, pH 7.0). The 3 mL reaction mixture contained 50 mM phosphate buffer (pH 7–8), 13 mM methionine, 75 mM NBT, 2 mM riboflavin (added at the last), 1 mM EDTA, and 50 μL of the enzyme extract, and the tubes were shaken and placed 30 cm below a light source consisting of two 15 W fluorescent lamps. The reaction was started by switching on the light and was allowed to run for 10 min during which time it was found earlier to be linear. The reaction was stopped by switching off the light and the tubes were covered with a black cloth. The absorbance by the reaction mixture at 560 nm was read. There was no measurable effect of the diffused room light. The reaction mixture lacking the enzyme developed the maximum color and this decreased with an increasing volume of the enzyme extract added. Log As60 was plotted as a function of the volume of the enzyme extract used in the reaction mixture.

#### 4.5.4. Phenylalanine Ammonia Lyase (PAL)

The enzyme extract was prepared following the protocol of Havir [38]. Borate buffer (0.5 mL), enzyme solution (0.2 mL), and water (1.3 mL) were mixed and the reaction was initiated by mixing 1 mL L-phenylalanine solution. It was then incubated at 32 °C for 60 min. Tri-choloroacetic acid (1 M, 0.5 mL) was added to stop the reaction. Further, absorbance was measured at 290 nm to assess the activity of phenyl alanine ammonia lyase using a UV–vis 1700 spectrophotometer.

#### 4.5.5. Polyphenol Oxidase (PPO)

Polyphenol oxidase activity was determined according to Gauillard et al. [39]. The enzyme extract was prepared by homogenizing 100 mg leaf sample in 2 mL 0.1 M phosphate buffer and centrifuging. Enzyme activity was measured by taking 1.4 mL of 0.1 M citrate–phosphate buffer in 0.5 mL TNB and 1 mL 2 mM catechol solution. The reaction was started by adding 100 µL of enzyme extract and absorbance was read at 412 nm in 30-s intervals for 3 min. The polyphenol oxidase activity was expressed by the change in absorbance (change in optical density; ΔOD) per min per mg fresh weight.

#### 4.5.6. Ascorbate Peroxidase (APx)

Ascorbate peroxidase (APX) activity in rice leaves was assessed as per the methods of Nakano and Asada [40]. In the 10 µL enzyme extract (prepared as describe for PPO), 180 µL 0.2 mM ascorbate and 0.2 mM hydrogen peroxide were added to start the reaction. The absorbance was recorded at 290 nm for 120 s using the spectrophotometer. APX activity was calculated according to Maksimovic and Zivanovic [41].

### 4.6. Biomass Accumulation

Root and shoot dry weight were recorded by uprooting the plants and gently washing the soil of the pots until clean in running tap water. Roots and shoots were detached. Fresh tissues were weighed and kept in a hot air oven at 60 °C ± 5 until a stable weight was achieved. The dry weight was recorded after drying of the samples.

### 4.7. Shoot and Root Length and Number of Tillers

Root and shoot length were measured at the maximum vegetative phase. Roots were carefully washed from the pots to harvest and measure. The number of tillers was counted at 90 DAS. The tillers bearing a panicle were counted as effective tillers and tillers without a panicle was counted as non-effective tillers.

### 4.8. Root Scanning

The roots were carefully washed out from the respective pots and cleaned twice with SDW. The roots were scanned in the EPSON Expression 12,000 XL scanner and different architecture parameters were recorded using the WINRHIZO Pro software. Parameters such as the root area, root volume, total length, number of tips, forks, fractal dimension, and average diameter were taken. The results were analyzed and presented in the text.

### 4.9. Gene Expression Study

The total plant RNA was isolated using the PureLink^TM^ RNA Mini Kit (Invitrogen, Waltham, MA, USA) following the manufacturer’s instructions. Total RNA was immediately converted to cDNA using the High Capacity RNA-to-cDNA kit (Thermo Fisher Scientific, Waltham, MA, USA). The yield of cDNA among different treatments of the two cultivars was assessed using a nanodrop and a final concentration of 50 ng/µL was used uniformly for assessing the differences in the transcript levels among the treatments. The cDNA was used for RT-PCR studies with key genes for salinity stress tolerance taking *Actin* as the endogenous control. The gene expression analyses were carried out using the RT-qPCR (Bio-Rad, Hercules, CA, USA) and Eva Green SYBR Green Supermix Kit (Bio-Rad, Hercules, CA, USA) in triplicate. Gene-specific primers (1.5 µL each; Appendix A) were taken at 10 pmol/µL in a reaction mixture of 10 µL along with 2 µL cDNA and 5 µL master mix. The RT-qPCR conditions were described elsewhere [6]. The data was normalized using the 2^−∆∆Ct^ method and to the Ct values of *Actin* as per Livak and Schmittgen [42].

### 4.10. CSLM Study for Na^+^ Accumulation in Leaf Tissues

In this study, relative Na^+^ accumulation in rice leaves was compared among different treatments. The fine sections of were made and treated with cell impermeant Sodium Green™, Tetra (Tetramethylammonium) Salt. The unbound dye was removed by washing with sterile distilled water (SDW). The slide was mounted on a grease-free glass slide and visualized under a 488 nm Laser in Confocal Scanning Laser Microscope (CSLM). The X–Y plane images were captured using a uniform camera setting and were processed in the Nikon NIS element software.

### 4.11. Statistical Analysis

The pot trial was conducted under a glass house condition with seven treatments and 3 replications of each (Figure 1) in a randomized complete block design (RCBD). The hypothesis was to evaluate microbial induction of stress tolerance in two different rice cultivars. This experiment was carried out to investigate the efficiency of plant rhizospheric microorganisms and endophytes to mitigate salinity stress in rice (*Oryza sativa*). Inoculation was done through seed treatment. Data were analyzed and means were compared with Duncan’s multiple range test at *p* ≤ 0.05.

## 5. Conclusions

Inoculation with halotolerant endophytes and rhizobacteria were found to modulate salt stress tolerance in rice. Inoculation was found to have positive role in improving oxidative enzymes, proline, chlorophyll, carotenoids, shoot length, root length, shoot dry weight, root dry weight, and number of effective tillers. In terms of plant biomass accumulation and the number of effective tillers, T4 (*Bacillus haynesii* 2P2) performed best. These two are key parameters to evaluate stress mitigation effects in the plants under salinity. In a generalized manner, enzyme activity was highest in T4, and gene expression was highest in T5 and T6 in CO51, whereas T6 and T7 were highest in the case of PB1. The core summary of different mechanisms indicated that T4 performed best amongst all, which could be combined as a consortium with T5 (*Bacillus safensis* BTL5) in CO51 and with T7 (*Pseudomonas fluorescens* 1001) in PB1. The third-best possible member of this consortium could be T6 (*Brevibacterium frigoritolerans* W19) in both CO51 and PB1. This conclusion of this study would be practically useful for forming a consortium based on plant responses to different rhizospheric and endophytic microbes. This could be a useful tool in evaluating microbial resources for climate-resilient agriculture. Further evaluation of these strains for consortial application and adaptability should be taken up.

## Figures and Tables

**Figure 1 plants-12-00976-f001:**
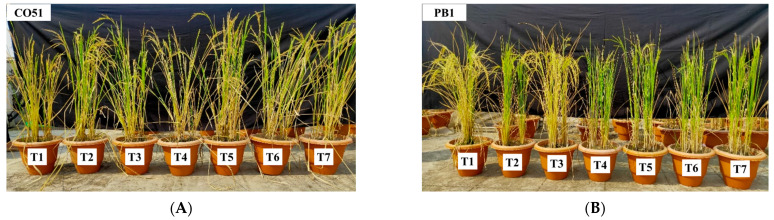
Effect of inoculation of halotolerant endophytes and rhizobacteria on two rice varieties (**A**) CO51, and (**B**) Pusa Basmati 1. Sequence of pots from left to right (in both of the figures): T1 = Negative control, T2 = Positive control (200 mM NaCl), T3 = 200 mM NaCl + *Trichoderma viride*, T4 = 200 mM NaCl + *Bacillus haynesii* 2P2, T5 = 200 mM NaCl + *Bacillus safensis* BTL5, T6 = 200 mM NaCl + *Brevibacterium frigoritolerans* W19, and T7 = 200 mM NaCl + *Pseudomonas fluorescens*.

**Figure 2 plants-12-00976-f002:**
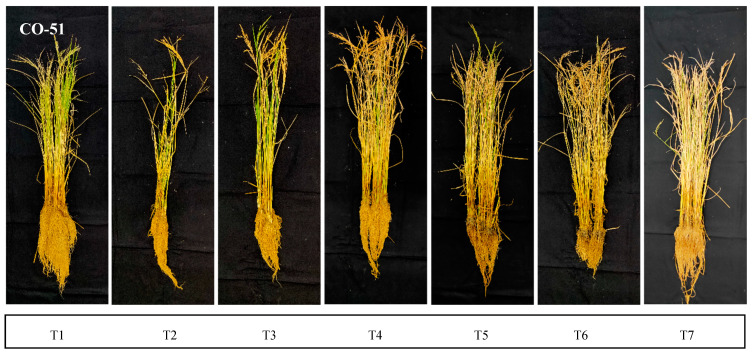
Effect of inoculation of halotolerant endophytes and rhizobacteria on root and shoot development in two rice varieties: CO51 and Pusa Basmati-1 (PB1). Sequence of pots from left to right: T1 = Negative control, T2 = Positive control (200 mM NaCl), T3 = 200 mM NaCl + *Trichoderma viride*, T4 = 200 mM NaCl + *Bacillus haynesii* 2P2, T5 = 200 mM NaCl + *Bacillus safensis* BTL5, T6 = 200 mM NaCl + *Brevibacterium frigoritolerans* W19, and T7 = 200 mM NaCl + *Pseudomonas fluorescens*.

**Figure 3 plants-12-00976-f003:**
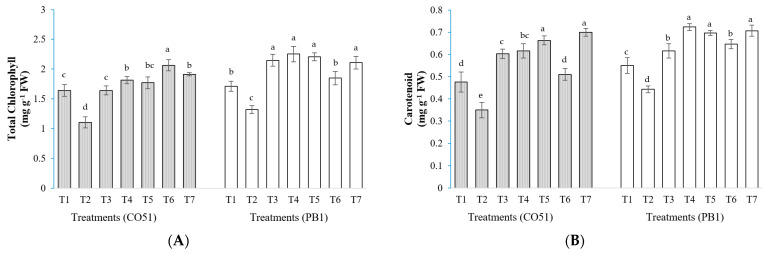
Effect of inoculation of halotolerant endophytes and rhizobacteria on photosynthetic pigments and compatible solute accumulation. (**A**) Chlorophyll content, (**B**) Carotene content, and (**C**) Proline accumulation in two rice varieties, CO51 and PB1. T1 = Negative control, T2 = Positive control (200 mM NaCl), T3 = 200 mM NaCl + *Trichoderma viride*, T4 = 200 mM NaCl + *Bacillus haynesii* 2P2, T5 = 200 mM NaCl + *Bacillus safensis* BTL5, T6 = 200 mM NaCl + *Brevibacterium frigoritolerans* W19, and T7 = 200 mM NaCl + *Pseudomonas fluorescens.* The treatment data with the same letters within the same cultivar are statistically non-significant at *p* ≤ 0.05. Means are separated by Duncan’s multiple range test. Error bars indicate standard deviation.

**Figure 4 plants-12-00976-f004:**
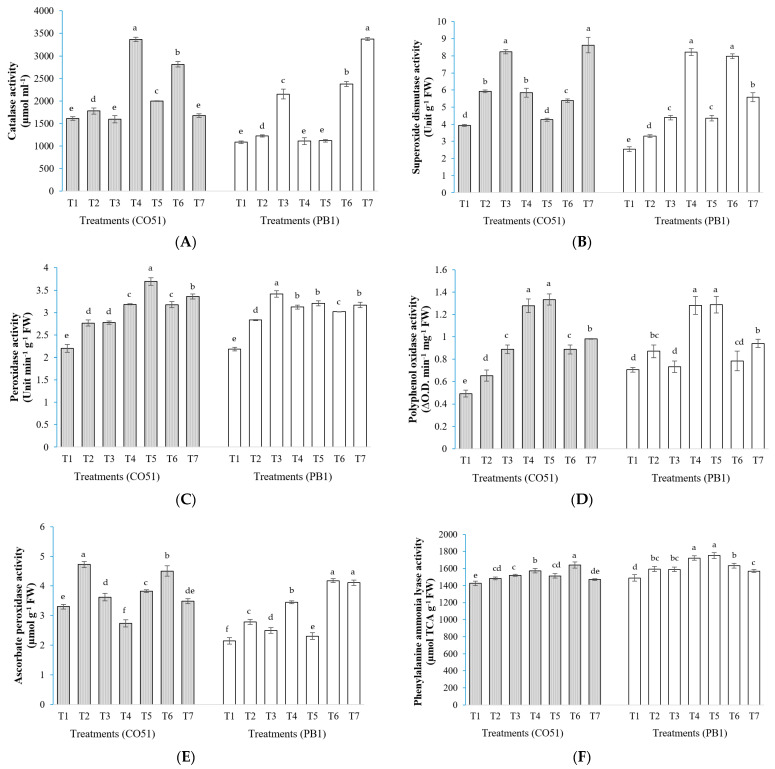
Effect of inoculation of halotolerant endophytes and rhizobacteria on the antioxidant enzyme activity in two rice varieties, CO51 and PB1. (**A**) Catalase, (**B**) Superoxide dismutase, (**C**) Peroxidase, (**D**) Polyphenol oxidase, (**E**) Ascorbate peroxidase, and (**F**) Phenylalanine ammonia lyase. T1 = Negative control, T2 = Positive control (200 mM NaCl), T3 = 200 mM NaCl + *Trichoderma viride*, T4 = 200 mM NaCl + *Bacillus haynesii* 2P2, T5 = 200 mM NaCl + *Bacillus safensis* BTL5, T6 = 200 mM NaCl + *Brevibacterium frigoritolerans* W19, and T7 = 200 mM NaCl + *Pseudomonas fluorescens.* The treatment data with the same letters within the same cultivar are statistically non-significant at *p* ≤ 0.05. Means are separated by Duncan’s multiple range test. Error bars indicate standard deviation.

**Figure 5 plants-12-00976-f005:**
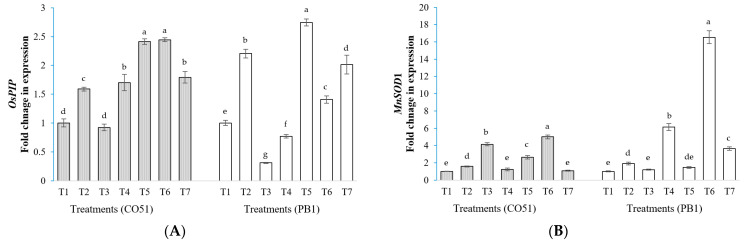
Relative expression of different salt responsive genes in the two rice varieties, CO51 and PB1. (**A**) *OsPIP*1, (**B**) *MnSOD*1, (**C**) *cAPXa*, (**D**) *CATa*, (**E**) *SERF*, and (**F**) *DHN*. T1 = Negative control, T2 = Positive control (200 mM NaCl), T3 = 200 mM NaCl + *Trichoderma viride*, T4 = 200 mM NaCl + *Bacillus haynesii* 2P2, T5 = 200 mM NaCl + *Bacillus safensis* BTL5, T6 = 200 mM NaCl + *Brevibacterium frigoritolerans* W19, and T7 = 200 mM NaCl + *Pseudomonas fluorescens;* The treatment data with the same letters within the same cultivar are statistically non-significant at *p* ≤ 0.05. Means are separated by Duncan’s multiple range test. Error bars indicate standard deviation.

**Figure 6 plants-12-00976-f006:**
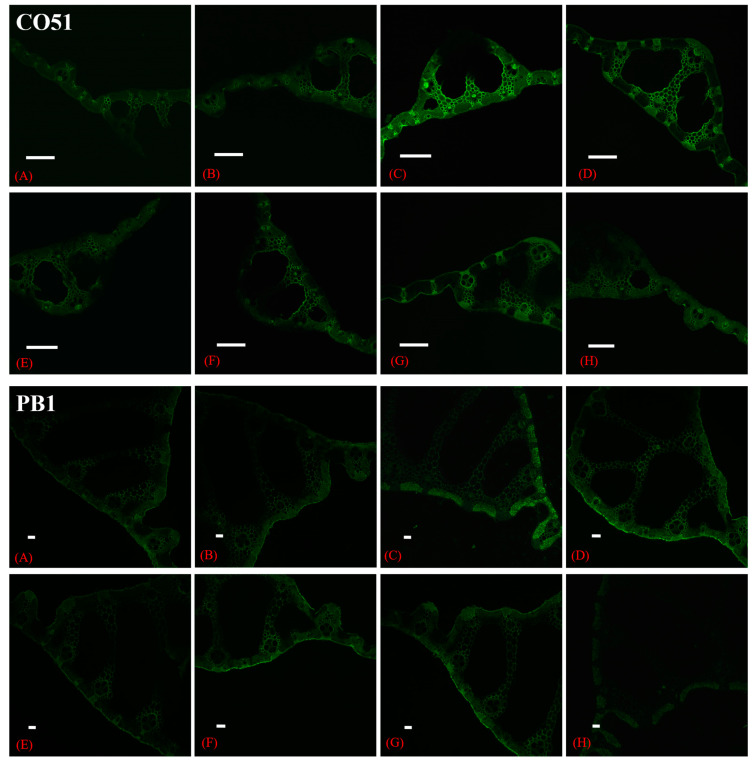
Confocal scanning laser microscopic images showing the difference in sodium accumulation in cross sections of CO51 and PB1 rice leaves. (**A**,**B**) Negative control, (**C**) Positive control (200 mM NaCl), (**D**) 200 mM NaCl + *Trichoderma viride*, (**E**) 200 mM NaCl + *Bacillus haynesii* 2P2, (**F**) 200 mM NaCl + *Bacillus safensis* BTL5, (**G**) 200 mM NaCl + *Brevibacterium frigoritolerans* W19, and (**H**) 200 mM NaCl + *Pseudomonas fluorescens*. Scale bar = 50 µm.

**Table 1 plants-12-00976-t001:** Effect of inoculation of halotolerant endophytes and rhizobacteria on growth parameters of two rice varieties, CO51 and PB1.

Treatments	Shoot Length (cm)		Root Length (cm)		Shoot Dry Weight (g)		Root Dry Weight (g)	
	CO51	PB1	CO51	PB1	CO51	PB1	CO51	PB1
T1	78.67 ^bc^ ± 1.15	68.33 ^b^ ± 1.53	34.33 ^a^ ± 1.15	37.33 ^b^ ± 0.58	6.63 ^d^ ± 0.26	7.39 ^e^ ± 0.14	6.03 ^bc^ ± 0.25	7.12 ^d^ ± 0.29
T2	75.67 ^c^ ± 1.53	73.50 ^a^ ± 3.04	22.67 ^c^ ± 2.08	31.33 ^e^ ± 0.58	5.11 ^e^ ± 0.25	6.00 ^f^ ± 0.2	2.99 ^d^ ± 0.52	4.41 ^e^ ± 0.24
T3	77.67 ^bc^ ± 1.53	71.67 ^ab^ ± 0.58	23.33 ^c^ ± 1.15	43.00 ^a^ ± 1.73	7.71 ^b^ ± 0.34	8.73 ^d^ ± 0.15	5.51 ^c^ ± 0.19	7.01 ^d^ ± 0.21
T4	80.67 ^b^ ± 1.53	69.33 ^b^ ± 1.15	32.13 ^a^ ± 1.21	34.33 ^cd^ ± 1.53	8.10 ^ab^ ± 0.13	11.01 ^a^ ± 0.17	6.60 ^a^ ± 0.14	7.99 ^ab^ ± 0.15
T5	80.67 ^b^ ± 1.15	73.67 ^a^ ± 1.53	28.33 ^b^ ± 1.53	34.83 ^cd^ ± 0.76	8.19 ^a^ ± 0.22	10.13 ^b^ ± 0.36	6.27 ^ab^ ± 0.3	7.61 ^bc^ ± 0.25
T6	84.00 ^a^ ± 2	74.00 ^a^ ± 2.65	22.33 ^c^ ± 1.53	36.33 ^bc^ ± 1.15	7.28 ^c^ ± 0.36	10.27 ^b^ ± 0.1	5.71 ^c^ ± 0.19	7.31 ^cd^ ± 0.21
T7	77.33 ^c^ ± 1.15	71.67 ^ab^ ± 2.31	33.67 ^a^ ± 0.58	33.67 ^d^ ± 1.53	7.27 ^c^ ± 0.22	9.59 ^c^ ± 0.38	6.26 ^ab^ ± 0.15	8.11 ^a^ ± 0.2

Note: The treatment data with the same letters are statistically non-significant at *p* ≤ 0.05. Means are separated by Duncan’s multiple range test. T1 = Negative control, T2 = Positive control (200 mM NaCl), T3 = 200 mM NaCl + *Trichoderma viride*, T4 = 200 mM NaCl + *Bacillus haynesii* 2P2, T5 = 200 mM NaCl + *Bacillus safensis* BTL5, T6 = 200 mM NaCl + *Brevibacterium frigoritolerans* W19, and T7 = 200 mM NaCl + *Pseudomonas fluorescens.*

**Table 2 plants-12-00976-t002:** Effect of inoculation of halotolerant endophytes and rhizobacteria on growth parameters of two rice varieties, CO51 and PB1.

Treatments	Number of Tillers		Number of Effective Tillers		Number of Non-Effective Tillers	
	CO51	PB1	CO51	PB1	CO51	PB1
T1	6.58 ^c^ ± 0.38	7.33 ^ab^ ± 0.29	6.17 ^c^ ± 0.29	5.42 ^ab^ ± 0.38	0.42 ^a^ ± 0.14	1.92 ^a^ ± 0.14
T2	3.92 ^e^ ± 0.25	5.62 ^e^ ± 0.27	3.67 ^e^ ± 0.25	3.95 ^d^ ± 0.21	0.25 ^a^ ± 0	1.67 ^a^ ± 0.29
T3	4.97 ^d^ ± 0.38	6.42 ^d^ ± 0.29	4.55 ^d^ ± 0.38	5.25 ^bc^ ± 0.5	0.42 ^a^ ± 0.14	1.17 ^b^ ± 0.29
T4	7.33 ^b^ ± 0.38	6.75 ^bcd^ ± 0.43	7.00 ^b^ ± 0.43	5.67 ^ab^ ± 0.38	0.33 ^a^ ± 0.29	1.08 ^b^ ± 0.29
T5	7.67 ^b^ ± 0.52	6.67 ^cd^ ± 0.14	7.42 ^b^ ± 0.72	5.83 ^ab^ ± 0.14	0.25 ^a^ ± 0.25	0.83 ^b^ ± 0.14
T6	7.58 ^b^ ± 0.52	7.25 ^abc^ ± 0.25	7.33 ^b^ ± 0.38	6.17 ^a^ ± 0.38	0.25 ^a^ ± 0.25	1.08 ^b^ ± 0.29
T7	8.92 ^a^ ± 0.52	7.75 ^a^ ± 0.43	8.50 ^a^ ± 0.66	4.33 ^d^ ± 0.76	0.42 ^a^ ± 0.29	1.17 ^b^ ± 0.29

Note: The treatment data with the same letters are statistically non-significant at *p* ≤ 0.05. Means are separated by Duncan’s multiple range test. T1 = Negative control, T2 = Positive control (200 mM NaCl), T3 = 200 mM NaCl + *Trichoderma viride*, T4 = 200 mM NaCl + *Bacillus haynesii* 2P2, T5 = 200 mM NaCl + *Bacillus safensis* BTL5, T6 = 200 mM NaCl + *Brevibacterium frigoritolerans* W19, and T7 = 200 mM NaCl + *Pseudomonas fluorescens.*

**Table 3 plants-12-00976-t003:** Effects of microbial inoculation on root architectural parameters.

Treatments	Length of Total Root (cm) *	Projection Area (cm^2^)	Surface Area (cm^2^)	Average Diameter (mm)	Root Volume (cm^3^)	Fractal Dimension	Tips	Forks
CO51								
T1	2861.81 ^a^ ± 125.42	428.92 ^a^ ± 9.74	1327.15 ^a^ ± 17.17	4.35 ^c^ ± 0.04	48.92 ^a^ ± 0.26	5.20 ^b^ ± 0.02	2883.33 ^c^ ± 93.54	12122.64 ^a^ ± 226.35
T2	1889.62 ^c^ ± 57.26	215.13 ^f^ ± 11.77	692.05 ^f^ ± 7.16	2.30 ^e^ ± 0.02	20.49 ^f^ ± 0.64	3.32 ^f^ ± 0.02	2583.00 ^d^ ± 23.3	6874.53 ^d^ ± 141.68
T3	2241.92 ^b^ ± 71.66	314.81 ^d^ ± 17.5	967.69 ^d^ ± 5.13	4.17 ^d^ ± 0.05	34.16 ^d^ ± 0.5	4.76 ^e^ ± 0.05	3281.00 ^b^ ± 72.64	7065.30 ^d^ ± 228.98
T4	2400.66 ^b^ ± 45.82	370.39 ^c^ ± 7.26	1185.51 ^c^ ± 11.08	4.75 ^b^ ± 0.03	46.75 ^b^ ± 1.45	5.24 ^b^ ± 0.04	1924.67 ^g^ ± 79.74	9689.01 ^c^ ± 47.54
T5	2042.90 ^c^ ± 172.17	280.42 ^e^ ± 15.47	855.92 ^e^ ± 46.37	5.31 ^a^ ± 0.03	48.38 ^ab^ ± 1.73	6.53 ^a^ ± 0.06	2424.00 ^e^ ± 56.51	5977.80 ^e^ ± 184.96
T6	1911.03 ^c^ ± 24.91	278.87 ^e^ ± 8.57	850.08 ^e^ ± 16.61	4.33 ^c^ ± 0.1	30.81 ^e^ ± 2.26	4.92 ^d^ ± 0.07	2256.33 ^f^ ± 49.05	4859.81 ^f^ ± 67.45
T7	2931.01 ^a^ ± 108.1	405.36 ^b^ ± 11.89	1249.99 ^b^ ± 42.18	4.08 ^d^ ± 0.06	43.13 ^c^ ± 1.08	5.06 ^c^ ± 0.08	3629.00 ^a^ ± 66.55	9982.93 ^b^ ± 230.85
PB1								
T1	1303.12 ^e^ ± 80.02	241.45 ^c^ ± 14.48	643.98 ^f^ ± 14.91	3.26 ^bc^ ± 0.34	28.85 ^c^ ± 1.11	2.74 ^c^ ± 0.02	1468.67 ^e^ ± 34.95	3991.03 ^d^ ± 129.79
T2	751.06 ^f^ ± 61.2	112.28 ^e^ ± 2.17	352.65 ^g^ ± 5.56	2.46 ^d^ ± 0.02	16.98 ^d^ ± 0.3	1.96 ^d^ ± 0.04	1178.00 ^f^ ± 15.52	2702.45 ^e^ ± 81.97
T3	1471.05 ^d^ ± 22.76	231.16 ^c^ ± 12.07	715.19 ^e^ ± 7.8	3.16 ^bc^ ± 0.05	31.26 ^bc^ ± 1.92	3.30 ^b^ ± 0.11	1492.00 ^e^ ± 25.71	5280.76 ^b^ ± 65.02
T4	1863.88 ^b^ ± 53.22	292.11 ^b^ ± 2.93	1047.92 ^b^ ± 11.26	3.38 ^b^ ± 0.02	35.80 ^a^ ± 1.71	3.59 ^b^ ± 0.12	2553.67 ^b^ ± 69.74	5202.17 ^b^ ± 238.94
T5	1747.94 ^c^ ± 37.06	206.25 ^d^ ± 4.01	926.43 ^c^ ± 6.8	3.17 ^bc^ ± 0.06	30.91 ^bc^ ± 1.45	3.37 ^b^ ± 0.02	1763.67 ^d^ ± 49.24	4861.84 ^c^ ± 120.06
T6	1427.39 ^d^ ± 70.47	236.92 ^c^ ± 8.84	759.30 ^d^ ± 22.91	3.03 ^c^ ± 0.08	32.24 ^b^ ± 0.87	3.42 ^b^ ± 0.3	2468.33 ^c^ ± 28.75	4811.04 ^c^ ± 99.7
T7	2052.01 ^a^ ± 48.85	312.63 ^a^ ± 7.44	1075.60 ^a^ ± 21.55	4.88 ^a^ ± 0.04	36.41 ^a^ ± 0.82	3.93 ^a^ ± 0.21	2623.67 ^a^ ± 26.1	5795.27 ^a^ ± 79.6

Note: Means with the same letters do not differ significantly at *p* ≤ 0.05. Means are separated by DMRT. T1 = Negative control, T2 = Positive control (200 mM NaCl), T3 = 200 mM NaCl + *Trichoderma viride*, T4 = 200 mM NaCl + *Bacillus haynesii* 2P2, T5 = 200 mM NaCl + *Bacillus safensis* BTL5, T6 = 200 mM NaCl + *Brevibacterium frigoritolerans* W19, and T7 = 200 mM NaCl + *Pseudomonas fluorescens.** Cumulative length of total root in the entire root system.

**Table 4 plants-12-00976-t004:** Treatment details used for the pot trial.

Sr. No.	Treatment	Details of the Treatments (Separately for Both Cultivars)
1.	T1	Negative control
2.	T2	Positive control (200 mM NaCl)
3.	T3	200 mM NaCl + *Trichoderma viride*
4.	T4	200 mM NaCl + *Bacillus haynesii* 2P2
5.	T5	200 mM NaCl + *Bacillus safensis* BTL5
6.	T6	200 mM NaCl + *Brevibacterium frigoritolerans* W19
7.	T7	200 mM NaCl + *Pseudomonas fluorescens* 1001

## Data Availability

Not applicable.

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
