# Peer review of "Salinity Alleviation and Reduction in Oxidative Stress by Endophytic and Rhizospheric Microbes in Two Rice Cultivars"

_plants, 2023, doi:10.3390/plants12050976_

Round 1
Reviewer 1 Report
The manuscript is well-written and well-organized.
However, a few minor corrections should be performed, e.g., no hypothesis and no statistical analyses for data presented in the figures (!). Differences between the two rice varieties Pusa Basmati-1 (PB1) and CO51 should be presented. The footnote of the Tables is unclear, i.e., "Means with same alphabet..." rather than letters (it should be improved + grammar). The abbreviation for the Duncan test - DMRT is not popular, use the full name.
The sentence "The treatments were: T1 = Negative 390 control, T2 = Positive control (200mM NaCl), T3 = 200m NaCl + Trichoderma viride, T4= 200mM NaCl + Bacillus haynesii 2P2, T5= 200mM NaCl + Bacillus safensis BTL5, T6= 200mM NaCl + Brevibacterium frigoritolerans W19, and T7= 200mM NaCl + Pseudomonas fluorescens 1001". should be added in the figure caption.
Author Response
Response to the comments of reviewer #1
Comments and Suggestions for Authors
Comment 1: The manuscript is well-written and well-organized. However, a few minor corrections should be performed, e.g.,---
Reply: The authors are grateful to the anonymous reviewer for their kind suggestions and corrections. We have made the changes to the manuscript based on the comments of all three esteemed reviewers. The response to queries is given here, and the corrections have been made in the concerned sections of the MS.
Sub-Comment (1a): No hypothesis and no statistical analyses for data presented in the figures (!).
Reply: In light of this comment, the data has been analysed based on Duncan's Multiple Range Test and the statistical significance at p≥0.05 added in the figure. Details are added in the section 4.11 of the manuscript.
Sub-Comment (1b): Differences between the two rice varieties Pusa Basmati-1 (PB1) and CO51 should be presented.
Reply: The graphs are now separated in two parts to show the differences between two rice varieties. A detail also added indicating varietal differences in last para of the Introduction.
Sub-Comment (1c): The footnote of the Tables is unclear, i.e., "Means with same alphabet..." rather than letters (it should be improved + grammar).
Reply: The footnote has been changed to “The treatment data having same alphabets, are statistically non-significant at p≤0.05 and means are separated by Duncan's multiple range test”.
Sub-Comment (1d): The abbreviation for the Duncan test - DMRT is not popular, use the full name.
Reply: Changed to “Duncan's multiple range test”
Comment 2: The sentence "The treatments were: T1= Negative control, T2= Positive control (200mM NaCl), T3= 200m NaCl + Trichoderma viride, T4= 200mM NaCl + Bacillus haynesii 2P2, T5= 200mM NaCl + Bacillus safensis BTL5, T6= 200mM NaCl + Brevibacterium frigoritolerans W19, and T7= 200mM NaCl + Pseudomonas fluorescens 1001". should be added in the figure caption.
Reply: The caption has been corrected in all the Figures and Tables of the manuscript.

Reviewer 2 Report
Abstract:
The abstract lacks a brief description of the relevance of the topic and lacks an objective. If there is no question, the reader does not know exactly why the work was done.
Introduction:
Ln 34-35 this work does not address the problems generated by climate change. Why does the manuscript start with this?
Literature reference is missing! ln 42-43 "Salinity stress in plants has severe detrimental effects on nutrient uptake, osmotic balance, membrane integrity and overall growth, thus hindering the overall crop dynamics of the plant.
Ln 54-55 "Microbes have been reported to have an enormous capacity to sustain plant growth under salinity [6]. Such as...?
Ln 66-67. ln "Both rhizosphere and endophytic microbes have common and non-common mechanisms for attenuating salinity, due to their niche [5]. Again....too general sentence, please define these mechanisms.
Ln 81 Please insert one more sentence about the ultimate goal/expected outcome why the authors decided to do this work.
Ln 390 Why CO51 and PB1 were selected as the experimenatl material? Please describe the varieties.
Ln 394-395 'Nursery was raised for these two cultivars and inoculation was done by seedling dip treatment following standard protocol.'
What standard protocol do you mean? Please explain.
Ln 396 Plants were raised following standard cultivation practices and the observations were taken time to time.
What standard cultivation practices do you mean? Please explain.
Ln 390-393 The treatments were: T1= Negative control, T2= Positive control (200mM NaCl), T3= 200m NaCl + Trichoderma viride, T4= 00mM NaCl + Bacillus haynesii 2P2, T5= 200mM NaCl + Bacillus safensis BTL5, T6= 200mM NaCl + Brevibacterium frigoritolerans W19, and T7= 200mM NaCl + Pseudomonas fluorescens 1001.
Please convert this into table and insert into the Mat and methods section.
Figure 1: What effect is the picture about? Growth?
It is not shown in the figure which treatment each plant belongs to. Please indicate
Table 3 is not in the correct format.
Ln 372-373 these are not enzyme systems, but enzymes which together form the antioxidant enzyme system!
Ln 381-382 'It shows that microbes have cultivar dependent functional behaviour' This is not the behaviour of microbes, but the response of the species.
Conclusions
The evaluation of the results lacks synthesising findings.
Which is the most effective treatment to counteract salt stress? This should definitely be described in the results. A summary figure of the mechanisms outlined above is missing.
Author Response
Response to the comments of reviewer #2
Authors are grateful to the anonymous reviewer for their kind inputs and time for our manuscript. We have made the changes in the manuscript based on the comments of all the three esteemed reviewers and the corrected revision is being submitted along with this document.
Comments and suggestions for authors
Comment 1: Abstract:
The abstract lacks a brief description of the relevance of the topic and lacks an objective. If there is no question, the reader does not know exactly why the work was done.
Reply: Thank you for the kind suggestion, the abstract has been modified to include background, hypothesis, purpose, methodology, results and conclusion. Elaborations were made in initial sentences and conclusion in the abstract.
Introduction:
Comment 2: Ln 34-35 this work does not address the problems generated by climate change. Why does the manuscript start with this?
Reply: The climate change has been added as opening statement of the Introduction section. Since the paper deals with abiotic stress (salinity), and the aggravation of the problem is majorly routed with climate change and related aspects. Thus, it was considered worthwhile to connect one of the major drivers of problem and to further narrow down introducing the problem in particular.
Comment 3: Literature reference is missing! ln 42-43 "Salinity stress in plants has severe detrimental effects on nutrient uptake, osmotic balance, membrane integrity and overall growth, thus hindering the overall crop dynamics of the plant.
Reply: Added reference [2]
Comment 4: Ln 54-55 "Microbes have been reported to have an enormous capacity to sustain plant growth under salinity [6]. Such as...?
Reply: Line added “such as improving nutrient uptake, osmotic balance, ionic balance, membrane stability, overall growth etc.”
Comment 5: Ln 66-67. ln "Both rhizosphere and endophytic microbes have common and non-common mechanisms for attenuating salinity, due to their niche [5]. Again....too general sentence, please define these mechanisms.
Reply: The sentence is modified to “Rhizosphere microbes act on the rhizosphere and could be instrumental in the plant-soil interface where plants encounter salinity, and endophytic microbes act inside the plant system where the ill effects of salinity are realized.”
Comment 6: Ln 81 Please insert one more sentence about the ultimate goal/expected outcome why the authors decided to do this work.
Reply: Sentence added “Looking at the increasing detrimental effects of salinity, it is important to characterize different microbial systems for staple food crops, such as rice. So that the base for effective climate-resilient cropping strategies could be widen and make Indian farming future-ready.”
Comment 7: Ln 390 Why CO51 and PB1 were selected as the experimenatl material? Please describe the varieties.
Reply: Portion added in the last para of Introduction “For our study, we have taken two cultivars with distinct features. The CO51, which is a short-duration, high-yielding rice cultivar, has a higher tolerance to stresses [10]. The second variety was Pusa Basmati 1 (PB1), which is the world’s first semi-dwarf Basmati variety, has higher yields, and is the most widely grown Basmati variety, but is relatively susceptible to some stresses [11].”
Comment 8:
Ln 394-395 'Nursery was raised for these two cultivars and inoculation was done by seedling dip treatment following standard protocol.'
What standard protocol do you mean? Please explain.
Reply: The reference of full protocol has been added and the details has been added “'Nursery was raised for these two cultivars as per protocol given in Nawaz et al. [31]. The microbial inoculation was done by seedling dip treatment. Seedlings were in respective culture broths (2mL per Liter) for a period of 30 minutes with 0.01% carboxy methyl cellulose as sticking agent. After a period of 1 hour, the seedlings were transplanted in the pots.
Comment 9:
Ln 396 Plants were raised following standard cultivation practices and the observations were taken time to time.
What standard cultivation practices do you mean? Please explain.
Reply: The details added in the material and methods section “In brief, rice plants were irrigated at field capacity, and 50% of nitrogen was applied in two split doses in a 30-day interval after inter-culture operation. Pots were randomized twice during the growth period to avoid any heterogeneity in light interception.”
Comment 10:
Ln 390-393 The treatments were: T1= Negative control, T2= Positive control (200mM NaCl), T3= 200m NaCl + Trichoderma viride, T4= 00mM NaCl + Bacillus haynesii 2P2, T5= 200mM NaCl + Bacillus safensis BTL5, T6= 200mM NaCl + Brevibacterium frigoritolerans W19, and T7= 200mM NaCl + Pseudomonas fluorescens 1001.
Please convert this into table and insert into the Mat and methods section.
Reply: The information has been added as Table 4 in Material and Methods section.
Comment 11:
Figure 1: What effect is the picture about? Growth?
It is not shown in the figure which treatment each plant belongs to. Please indicate
Reply: This Figure represents overall image of the experiments showing growth attribute of different treatments in one frame. The treatments are now labeled in image itself.
Comment 12:
Table 3 is not in the correct format.
Reply: The Table 3 indicated eight different root architectural parameters in two cultivars and we have tried to incorporate this complicated data with statistical analysis along with the standard deviation values. This altogether made a heavy table, which is formatted to shortest possible table format. However, based the comments of the reviewer, we are putting request to the MDPI copy editor to further improve the table formatting.
Comment 13:
Ln 372-373 these are not enzyme systems, but enzymes which together form the antioxidant enzyme system!
Reply: Corrected to “enzymes”
Comment 14:
Ln 381-382 'It shows that microbes have cultivar dependent functional behaviour' This is not the behaviour of microbes, but the response of the species.
Reply: Corrected as “It shows that both cultivars respond differently to the microbial inoculation. As seen from the findings of Sahu et al. [9] where differential microbial functions has been reported from two different cultivars”.
Comment 15: Conclusions
The evaluation of the results lacks synthesising findings.
Which is the most effective treatment to counteract salt stress? This should definitely be described in the results. A summary figure of the mechanisms outlined above is missing.
Reply: The conclusion section has been elaborated in light of this comment. The summarized effects of different treatments via different mechanisms has been added as “In terms of plant biomass accumulation and the number of effective tillers, T4 (Bacillus haynesii 2P2) performed best. These two are key parameters to evaluate stress mitigation effects in the plants under salinity. In a generalized manner, enzyme activity was highest in T4, gene expression was highest in T5 and T6 in CO51, whereas T6 and T7 were highest in the case of PB1. The core summary of different mechanisms indicated that T4 performed best amongst all, which could be combined as a consortium with T5 (Bacillus safensis BTL5) in CO51 and with T7 (Pseudomonas fluorescens 1001) in PB1. The third-best possible member of this consortium could be T6 (Brevibacterium frigoritolerans W19) in both CO51 and PB1.”

Reviewer 3 Report
Dear Editors,
Thank you so much for choosing me as a reviewer of the manuscript NASB-D-22-00522R1 entitled: “Salinity alleviation and reduction in oxidative stress by endo-phytic and rhizospheric microbes in two rice cultivars” submitted to Plants. I hope that my comments will help Authors to improve their manuscript.
Detailed remarks concerning the manuscript.
Key words. It is not recommended to use the same words or phrases that was used in the title manuscript. Please do needed changes.
I suggest to include in abstract one or two backroad sentences, hypothesis and purpose of the report, one or two sentences for methodology, a few sentences for results and one or two conclusion sentences.
Not only the clear purpose of the report but also the scientific hypothesis with the clear answer for the question stated as scientific hypothesis should be given
Please give the information concerning the bars on the figures. There is no information to what are referred to (SE or SD).
If it is possible please improve the quality of the Figure 6.
I suggest to divide discussion into the subsections similar to the subsections for results section.
Please give the practical application of the study.
The reference list should be prepared strictly according to the one scheme given guides for authors. There are many editorial mistakes. There is not possible to mention all of them. There are only some example: (i) once only the first word in the manuscript title is written with capital letter but the other time each word in the manuscript title is written with the capital letter, (ii) once the full, but the other time abbreviated title of the manuscript is presented, (iii) once “and” is between the last author and year but the other time not. Please go very carefully through the whole bibliography and do needed changes.
Author Response
Response to the comments of reviewer #3
Authors are grateful to the anonymous reviewer for valuable suggestions to improve the manuscript. We have made the changes in the manuscript based on the comments of all the three esteemed reviewers and the corrected revision is being submitted along with this document.
Comments to the Author
Detailed remarks concerning the manuscript.
Comment 1: Key words. It is not recommended to use the same words or phrases that was used in the title manuscript. Please do needed changes.
Reply: The suggestion is welcomed, however, the current keywords are designed keeping in the view the specific guidelines given by the journal (https://www.mdpi.com/journal/plants/instructions)
Comment 2: I suggest to include in abstract one or two backroad sentences, hypothesis and purpose of the report, one or two sentences for methodology, a few sentences for results and one or two conclusion sentences.
Not only the clear purpose of the report but also the scientific hypothesis with the clear answer for the question stated as scientific hypothesis should be given
Reply: The abstract has been modified to include background, hypothesis, purpose, methodology, results and conclusion. Elaborations were made in initial sentences and conclusion in the abstract.
Comment 3: Please give the information concerning the bars on the figures. There is no information to what are referred to (SE or SD).
Reply: It indicates standard deviation. Information has been added in the figure captions.
Comment 4: If it is possible please improve the quality of the Figure 6.
Reply: We have original large sized images which we are separately sending to the journal for final incorporation. This could further improve the clarity of the images.
Comment 5: I suggest to divide discussion into the subsections similar to the subsections for results section.
Reply: In the discussion, different mechanisms are compiled to support one result, in this case one study results are discussed at multiple places in discussion, and they are not distinctly arranged in the discussion. For e.g., the reduce accumulation of Na+ in leaves has different determinants, here we need to discuss results of enzyme activity, root architecture, gene expression, proline accumulation; it would be very difficult to segregate the interconnected sections. Therefore, discussion has been made in a single section.
Comment 6: Please give the practical application of the study.
Reply: The practical utility has been added in the conclusion section. Portion added “This conclusion of this study would be practically useful for forming consortium based on plant responses to different rhizospheric and endophytic microbes. This could be a useful tool in evaluating microbial resources for climate-resilient agriculture.”
Comment 7: The reference list should be prepared strictly according to the one scheme given guides for authors. There are many editorial mistakes. There is not possible to mention all of them. There are only some example: (i) once only the first word in the manuscript title is written with capital letter but the other time each word in the manuscript title is written with the capital letter, (ii) once the full, but the other time abbreviated title of the manuscript is presented, (iii) once “and” is between the last author and year but the other time not. Please go very carefully through the whole bibliography and do needed changes.
Reply: Based on the kind comments, the references has been corrected as per the format sent by the editorial office.

Round 2
